# VPNSniffer: Identifying VPN Servers Through Graph-Represented Behaviors

Submission Id: 1393

## ABSTRACT

Identifying VPN servers is a crucial task in various situations, such as geo-fraud detection, bot traffic analysis and network attack identification. Although numerous studies that focus on network traffic detection have achieved excellent performance in closed-world scenarios, particularly those methods based on deep learning, they may exhibit significant performance degradation due to changes in the network environment. To mitigate this issue, a few studies have attempted to use methods based on active probing to detect VPN servers. However, these methods still have some limitations. They cannot handle situations where probing responses are absent, and lack generalization due to their focus on specific VPNs. In this work, we propose VPNSniffer, which utilizes the graph-represented behaviors to detect VPN servers in real-world scenarios. VPNSniffer outperforms existing methods in four offline datasets. The results based on our datasets, which contain multiple different VPNs, also indicate that VPNSniffer has better generalization. Furthermore, we deploy VPNSniffer in an Internet Service Provider's (ISP) environment to evaluate its effectiveness. The results show that VPNSniffer can improve the coverage of sophisticated detection engines and serve as a complement to existing methods.

## CCS CONCEPTS

• **Security**; • **Information systems** → **Data Mining**; • **Computing methodologies** → **Artificial intelligence**;

## KEYWORDS

VPN Detection, Active Probing, Node Classification

**ACM Reference Format:**

Anonymous Author(s). 2018. VPNSniffer: Identifying VPN Servers Through Graph-Represented Behaviors. In *Proceedings of Make sure to enter the correct conference title from your rights confirmation emai (WWW'2024)*. ACM, New York, NY, USA, 10 pages. https://doi.org/XXXXXXX.XXXXXXX

## 1 INTRODUCTION

Due to the growing demand for privacy protection, VPNs have become increasingly popular tools [7]. Specifically, VPNs can encrypt users' network traffic and even hide users' identities to ensure communication security. However, VPNs may also be exploited to conduct geo-fraud, network attacks, malicious crawling, among

other abusive situations [14, 15, 33, 36]. For example, a website may identify an incoming IP address as located in Los Angeles, while the client behind this IP may actually be located in New York. Many websites (e.g., Netflix, ChatGPT) restrict incoming IPs to protect their commercial interests. Nevertheless, VPNs can help clients bypass these copyright protection strategies. Consequently, detecting VPN servers to prevent abusive activities is necessary.

In the field of VPN server detection, much research focuses on network traffic detection. Benefiting from advances in artificial intelligence technology, many researchers utilize machine learning and deep learning algorithms to identify VPN servers from network traffic. In particular, those deep learning-based methods exhibit high performance in lab environments [16, 21, 43, 48, 49]. Xie et al. [42] demonstrate that these methods might exhibit significant performance degradation when the network environment differs from the training scenarios. Their experiments indicate that some methods achieve an F1-score higher than 0.99 in the training environment, but the F1 score can fall below 0.40 in other testing situations. The sophisticated real-world network environment [2, 28, 32, 46, 47] may contain packet loss, network delay, partial network failures, and network updates. All of these network phenomena could alter traffic features, resulting in performance degradation for these traffic-based methods. The significant performance loss caused by the network environment inspires us to explore a method that does not rely on any network traffic payload information.

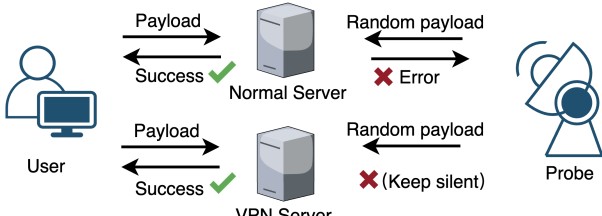

**Figure 1: Active Probing Illustration**

Recently, several works [4, 5, 10] have begun to detect VPN servers based on active probing, which are unaffected by the testing network environment. Specifically, researchers send probes to a target server and determine whether it is a VPN server based on the response information. Owing to different configuration strategies, responses from VPN servers and normal servers are different [5]. For instance, a normal server might return error information and close the connection after receiving an unexpected request packet, while VPN servers may remain silent until timeout, as shown in Figure 1. While previous methods based on active probing have promoted the development of VPN servers detection, they still exhibit two critical limitations. First, we discover that 16.44% of servers in our dataset do not respond with any information, and

previous work cannot handle this situation. Secondly, they only focus on certain specific VPNs, which may lead to performance degradation when applied to other VPNs.

To overcome these limitations, we utilize graph-represented behaviors to detect VPN servers. Our design is based on the following considerations: i) Compared to VPN servers, normal servers are accessed by numerous clients and typically exhibit a more sophisticated connection relationship. We construct a communication graph to capture servers' connection behaviors. This graph relies on communication information and can detect VPN servers that lack probing response information. ii) A client might access multiple VPN servers in a short time, and these servers may share the same configuration strategy and show similar probing responses. We construct a probing graph based on this phenomenon and utilize some new features related to probing response behaviors to capture the general characteristics of VPN servers and enhance the method's generalization.

Experimental results on four offline datasets demonstrate that our method achieves state-of-the-art (SOTA) performance. Our VPN datasets include at least 43 VPNs, and experimental results show that our method has better generalization. This paper aims to detect VPN servers in a real-world context. Therefore, we deploy our model in an ISP's environment and compare it with industrial detection engines. The results show that our model can assist these engines in identifying a greater number of VPN servers.

In summary, the contributions of this work are:

**New Observation:** We display some new observations regarding VPN servers, such as 'Stealth Port'. Based on these observations, we introduce several new features, including response types and port distribution, which can assist the security community in identifying abused VPN servers.

**New Technique:** We present VPNSniffer, which detects VPN servers using graph-represented behaviors. VPNSniffer not only exhibits better generalization but also outperforms previous methods and can enhance the coverage of sophisticated detection engines.

**New Paradigm:** To the best of our knowledge, we are the first to combine the features of active probing and node communication relationship to detect VPN servers, offering a unique perspective on VPN server detection.

**Ethics and piracy:** The offline dataset in this paper is derived from the real world. To mitigate potential piracy and ethical risks, our ISP partner anonymizes all client IPs and provides us only with limited information, such as ports and server IPs. We cannot obtain the payload of any raw traffic. Considering that a smaller dataset corresponds to a lower privacy risk, we establish a simulation environment in our laboratory to determine the optimal data collection scope. Our dataset was collected through our ISP partner during one hour on June 7, 2023. For online experiments, our system only receives information from a copy of the raw traffic, ensuring that if our model malfunctions, the network service remains unaffected.

## 2 BACKGROUND

We refer to the communication behaviors exposed in an ISP's environment as communication behaviors, while the probing response behaviors are termed probing response behaviors. Given that our dataset is collected from an ISP's environment and certain features

are derived from active probing, a description of the passive and active behaviors of VPNs can aid readers in understanding our motivation. Finally, we introduce the detection challenges in the real world.

**Communication Behaviors:** During VPN communication, a VPN client typically exhibits the following behaviors: i) Authentication: The client might need to send login information to the server for authentication, especially when utilizing commercial VPNs, such as TorGuard VPN. ii) Fast-connection: To optimize network performance, the VPN (e.g., Cryptostorm VPN) client may attempt to connect to several servers across various locations and select the one with the shortest response time. Subsequently, the VPN client establishes a secure communication tunnel with the selected server. iii) Keep-Alive: To prevent traffic leakage [29], the VPN (e.g., NordVPN) client may continuously send packets to ensure the server remains active. Consequently, when users communicate via the VPN tunnel, they might connect to multiple VPN servers. In an ISP's environment, if a user is observed accessing one VPN server, additional VPN servers often emerge within a short time window in the user's server access sequence. The VPN servers in this sequence might belong to the same vendor and share identical configuration strategies.

**Probing Response Behaviors:** Active probing [25] is commonly utilized to identify VPN servers. When researchers send their meticulously designed probes to the servers, VPN servers usually exhibit one of three response behaviors: i) Keep Silent: If the request lacks authentication information, the VPN server remains silent and does not respond. For servers that use the UDP protocol, they might not send any message. Those using the TCP protocol may close the connection and send a FIN/RST packet [10]. ii) Expected Response: Servers reply with content that matches the standard VPN protocol. For instance, when researchers send an OpenVPN request to a VPN server and receive a byte sequence that matches the response format [24], they can infer that this server is an OpenVPN server. iii) Unexpected Response: This could include various error messages, non-standard protocol responses.

**Detection Challenges** Owing to IP-sharing access technologies, such as Network Address Translation (NAT) [31], multiple endpoints might be behind a client IP. Due to the Proxy Auto-Configuration (PAC) mechanism [50], VPN software allows only a portion of application traffic to be routed through the VPN tunnel, while the traffic from other applications communicates outside the VPN tunnel. As a result, when a client IP communicates via a VPN, it does not only communicate with VPN servers but may also interact with normal servers. This complex environment presents challenges for identification. To mitigate this issue, we roughly remove certain abnormal client IPs during the dataset collection process, instead of assuming that the environment is pure.

## 3 DESIGN OF VPNSNIFFER

### 3.1 High Level Description of VPNSniffer

VPNSniffer is designed to identify VPN servers in real-world scenarios, which are more sophisticated than a lab environment. Given a detection time window $t$, we use $s^t_{(i,*)}$ to represent the set of servers accessed by user $i$. There are $n$ users who access servers within a time window of $t$. VPNSniffer aims to identify VPN servers from

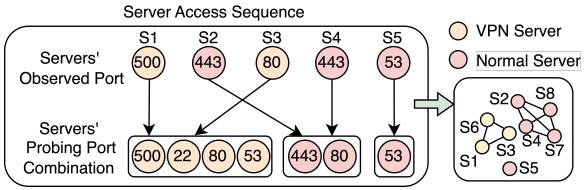

**Figure 2: VPNSniffer Model Architecture**

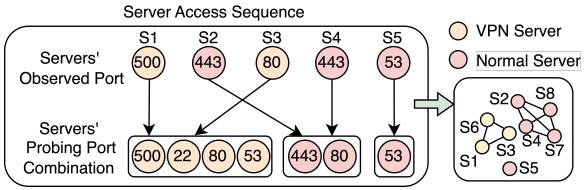

**Figure 3: Motivation of Probing Graph (S means server)**

this set $S$.

$$S = \sum_i^n s_{(i,\ vpn\ servers)}^t \cup \sum_i^n s_{(i,\ normal\ servers)}^t \quad (|s_{(i,*)}^t| >= 0) \quad (1)$$

We observe that the VPN servers' probing information and communication relationship may differ from that of normal servers. For instance, normal servers might respond with error messages after receiving a TCP probe with a random payload. However, VPN servers might directly close the connection. In terms of communication relationship, normal servers are typically accessed by a myriad of clients, leading to more sophisticated connection relationships compared to VPN servers. Therefore, we utilize both the probing graph and the communication graph to capture these features and detect VPN servers.

The architecture of VPNSniffer is depicted in Figure 2. We first construct the probing and communication graphs, embedding relevant information within their nodes. Subsequently, we encode each graph, concatenate their features, and use a classifier to identify the VPN servers. Finally, we deploy our system in the real world.

## 3.2 Probing Graph Construction

Ports are widely used to reveal the services supported by servers. Given a detection time window $t$, we refer to the port used by each server in a client's access server sequence as a *Observed Port* (as illustrated in Figure 3). Servers might provide various services using different ports. However, the clients' server access sequences only show partial ports of a server, so collecting only the ports from the *Observed Port* might lose some information. To this end, we conduct port scanning for each server to supplement more information. Specifically, we send TCP SYN and UDP packets to each server, which is a common method [24]. If we receive any TCP or UDP response, we determine that the port is open. We term the server's ports obtained from active probing as *Probing Port Combination*.

Contrary to our intuition, the *Probing Port Combination* of many VPN servers are not random. We roughly assume that if a port combination appears fewer than 10 times, it may be random. Such combinations account for only 18.47% of the total. Furthermore, we discover that servers belonging to the same vendor may have similar Probing Port Combinations. This might be because servers from the same provider often share a deployment strategy. For example, Psiphon3 VPN servers typically use the *Probing Port Combination* {443, 53, 22}, {443, 554, 22}, and so on. For more details, please refer to Appendix A. This phenomenon inspires us to construct a graph using this similarity.

Give the time window $t$, we assume that the server $s_j$ and $s_k$ in the set $s_{(i,*)}^t$ use similar *Poring Port Combination* $\mathcal{P}(s_j)$ and $\mathcal{P}(s_k)$, indicating they offer similar services. This similarity $\zeta$ is computed by using formula $\mathcal{J}$. We use $\mathcal{PG}_t$ to describe this relationship, where $\mathcal{E}^t$ and $\mathcal{D}^t$ represent the edge and node sets, respectively, and $\mathcal{N}(m)$ signifies the set of nodes that connect to node $m$. Please note that not every server has a connected node. To facilitate subsequent computations, we add self-loops for such isolated nodes. The node features of this graph are mainly derived from the active probing, therefore we refer to $\mathcal{PG}_t$ as the probing graph.

$$\mathcal{E}_i^t = \left\{(s_j,\ s_k)|\ s_j,\ s_k \in s_{(i,*)}^t\ and\ \mathcal{J}(\mathcal{P}(s_j)\ ,\ \mathcal{P}(s_k)) >= \zeta\right\} \quad (2)$$

$$\mathcal{J}(\mathcal{P}(s_j)\ ,\ \mathcal{P}(s_k)) = \frac{|\mathcal{P}(s_j)| \cap |\mathcal{P}(s_k)|}{|\mathcal{P}(s_j)| \cup |\mathcal{P}(s_k)|} \quad (3)$$

$$\mathcal{E}^t = \bigcup_{i=1}^n \mathcal{E}_i^t \qquad \mathcal{D}^t = \bigcup_{i=1}^n s_{(i,*)}^t \quad (4)$$

$$\mathcal{E}_{self-loop} = \left\{(m,\ m)\ |\ m \in \mathcal{D}^t\ and\ m \notin \mathcal{N}(m)\right\} \quad (5)$$

$$\mathcal{PG}_t = <\mathcal{D}^t,\ \mathcal{E}^t \cup \mathcal{E}_{self-loop}> \quad (6)$$

Our probes include application layer probes and transport layer probes. From the application layer probes, we extract features related to response types. Based on the transport layer probes, we derive features concerning response time, response length, termination state, and port distribution.

**Application Layer Probes:** Application layer probes include both traditional VPN protocol probes and popular protocol probes. Our design is based on the following observation: i) Many VPN servers still utilize traditional protocols [25], such as PPTP, IPSec, SSTP, and OpenVPN. In our dataset, at least 27.15% of VPN servers deploy these traditional VPN protocols. Consequently, we believe that traditional VPN protocols probes can help us discover VPN servers. ii) To bypass detection, many VPNs disguise their traffic to popular

protocols and use the corresponding standard ports [30, 34, 38]. For instance, they may mimic TLS traffic and use port 443. Since these servers do not actually offer TLS services, their response behaviors might differ from those of normal servers when we send a TLS request packet to them.

Specifically, our traditional VPN protocol probes and popular protocol probes encompass PPTP, IPSec, SSTP, OpenVPN, HTTP, TLS, SSH, DNS, and FTP. We employ these probes to obtain response types for each server. To accelerate probing speed, all of the above probes are sent to standard ports only. An overview of probe content is shown in Table 1. The response types reveal not only whether the server supports the corresponding protocol but also the server's deployment strategy. Below are two examples of probes and we refer interested readers to our code [1] for more details.

**OpenVPN Response Type:** We send a TCP packet containing \x38{8}\x00\x00\x00\x00 to a target server. If the server responds with \x00{1}\x40{*}\x00\x00\x00\x00, it strongly suggests that the server is running an OpenVPN service [25] since this response content matches the standard OpenVPN format. We set this response type as $\mathcal{RT}_o = 1$ ($o$ means OpenVPN). Previous work [5, 10] has shown that VPN servers may employ configurations to bypass probing attempts. We observe that VPN servers might also have the following response types after they receiving an OpenVPN probe: i) The server directly rejects the TCP connection, indicated by $\mathcal{RT}_o = 2$. ii) The server remains silent until a timeout occurs, which is represented as $\mathcal{RT}_o = 3$. iii) The server responds with an empty packet after establishing a TCP connection, denoted by $\mathcal{RT}_o = 4$. We observe that while many servers utilize port 1194 for communication in the real world, only 3.76% of those return a standard OpenVPN protocol response. This suggests that many VPN servers adopt probe-resistant strategies and a detailed analysis of response type is necessary. Furthermore, we discover that some normal servers use port 1194 to provide different services, like HTTP, we set these response type as $\mathcal{RT}_o = 5$. All of these response types enable us to understand the server in greater detail than simply focusing on whether a response is received.

Although popular probing tools like Nmap are also capable of probing VPN protocols, existing research [19, 22] suggest that servers might filter their probes due to exposed probing fingerprints. Therefore, we design our probes by a public python library [9] instead of using existing probing tools. Our probing result indicates that our VPN protocol probes have a response success rate about twice that of Nmap.

**DNS Response Type:** The DNS protocol allows clients to assign a DNS resolver to query the IP addresses of domains. We consider that the response behaviors of VPNs that mimic DNS traffic and communicate using port 53 may differ from normal VPNs. To distinguish between VPN and normal servers, we assign the target server as a DNS resolver and send a DNS query about google.com (a common test domain). The response types of servers include: i) The server answers with the IPs of google.com, indicating it provides DNS service, denoted as $\mathcal{RT}_d = 1$ ($d$ means DNS). ii) The server responds with All nameservers failed to answer, represented as $\mathcal{RT}d = 2$. Our intuition is that google.com, being one of the most globally popular domains, should be cached in a

DNS resolver. We discover that 34.24% of responsive VPN servers fail to answer, while only 18.89% of responsive normal servers provide that response. This difference can help us detect VPN servers. iii) The server does not reply with any response until a timeout occurs, referred as $\mathcal{RT}_d = 3$. iv) We also consider the response The DNS query name does not exist ($\mathcal{RT}_d = 4$) which also indicate this may be a fake DNS resolver. All of these response types can reveal a server's configuration for port 53.

In general, we utilize these application layer probes to gather response type features for each target server. These features can disclose server configurations and aid us in detecting VPN servers.

**Table 1: Summary of Application Layer Probes**

| Name | UDP/TCP | Probe Content | Port |
|---|---|---|---|
| SSTP | TCP | \x53\x53\x54\x50\x5F... | 443 |
| IPSec | UDP | {8}\x01\x10... | 500 |
| OpenVPN | TCP&UCP | \x38{8}\x00\x00\x00\x00 | 1194 |
| PPTP | TCP | \x00\x9c\{6}\x00\x01... | 1723 |
| FTP | TCP | Python TCP Connection | 21 |
| SSH | TCP | Python TCP Connection | 22 |
| DNS | UCP | Query google.com | 53 |
| HTTP | TCP | GET / HTTP/1.1\r\n... | 80 |
| TLS | TCP | Python SSL Connection | 443 |

**Transport Layer Probes:** The transport layer probes encompass both TCP and UDP probes. We extract the features of response time, response length, terminate state and port distribution based on these probes.

**Response Time and Length:** As defined earlier, refer to the port observed in the ISP's environment as *Observed Port*. We probe each *Observed Port* using TCP and UDP probes with 1500 bytes (standard Maximum Transmission Unit (MTU)) to collect the total packet length ($\mathcal{PL}$), packet count ($\mathcal{PC}$) and time duration ($\mathcal{DT}$, calculated as the difference between the last and first response times). To accelerate the probing process, we set the maximum timeout period to 300 seconds. For servers with multiple *Observed Ports*, we select results that exhibit the maximum values for $\mathcal{PL}$, $\mathcal{PC}$ and $\mathcal{DT}$. We observe that these features may differ between VPN and normal servers. For instance, VPN servers prefer to reply with a longer response time and a shorter response packet after receiving a TCP probe, as shown in Figure 4 and Figure 5. We believe this may be a strategy for VPNs to reduce the exposure risk.

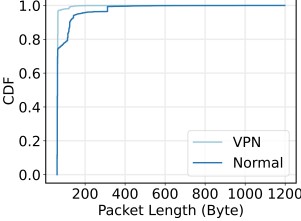
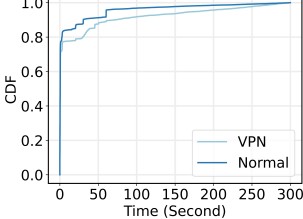

**Figure 4: Response Length**   **Figure 5: Response Time**

**Terminate State:** TCP servers might terminate a connection if they receive bytes that cannot be parsed. Previous research [10] has

indicated that receiving bytes beyond a certain threshold can lead a server to send a FIN packet, while a Linux server that closes a TCP connection with unread bytes in its buffer might trigger an RST packet. The RST/FIN threshold represents the minimum byte count that triggers an RST/FIN packet. Existing work [5] has shown that the RST threshold distribution differs between VPNs and normal servers. Therefore, while some servers might close the connection and not respond with content to avoid discovery, their termination behavior can also be utilized to detect VPN servers.

We consider that the RST/FIN threshold is influenced by a server's configuration strategy. Drawing from prior work [4, 5, 10], we utilize the RST and FIN thresholds as features to detect VPN servers. We randomly send TCP probes to estimate these thresholds for each server. We assume the value of RST/FIN threshold is smaller than the standard MTU. Notably, while the MTU is 1500 bytes, it's unnecessary to send 1500 probes. We can use the binary search method to optimize our probing. The scope of our probe contains *Observed Ports* and the ports of TCP services enumerated in Table 1. To ensure consistent feature dimensions and facilitate subsequent calculations, we implement the following operations: i) For servers with multiple *Observed Port* entries, we randomly select one with the smallest RST/FIN threshold. ii) If the *Observed Port* matches the ports of TCP services in Table 1, we also record its value.

**Port Distribution:** We perform port scanning for every server and refer to the ports obtained by probing as *Probing port*. We discover that servers may use multiple ports to communicate with clients in the ISP's environment. However, some of these ports do not respond to any message after receiving our probe, not even with a message to close the connection. We compute the difference in ports ($\mathcal{DP}$, $\mathcal{DP} = Observed\ Port - Probing\ Port$) to describe this phenomenon, which we refer to as 'Stealth Port'. In other words, $\mathcal{DP}$ means the port provides service for users but prohibits probing. In our dataset, the average $\mathcal{DP}$ of VPN servers is about 2.5 times that of normal servers. This difference may be due to some VPN servers adopting a more strict probe-resistant strategy. Additionally, we also compute the length of *Observed Port* and $|Observed\ Port| \cup |Probing\ Port|$ as features to detect VPN servers.

The port distribution sequence of servers may reveal the services supported by the server. However, our dataset indicates that approximately 90% of servers utilize fewer than 35 ports, and such sparse distribution (35/65535) poses a challenge to feature representation. We design the following formula to compress the port distribution into a 10-dimensional feature vector $\mathcal{V}$. The intention of this formula is to retain some information about the server's port patterns while compressing the features. Where $I$ is an indicator function, $p_j$ is port $j$ from *Observed Port* $\cup$ *Probing Port*. The indicator function takes the value 1 when two elements are the same and 0 when they are different.

$$\mathcal{V} = \mathcal{V}_0 \oplus \mathcal{V}_1 \oplus ... \oplus \mathcal{V}_9 \quad (7)$$

$$\mathcal{V}_i = \sum_{j=1}^{m} \mathcal{I}(p_j \bmod 10, i) \quad (8)$$

Generally, we obtain features derived from response types, response time, response length, terminate state, and port distribution for every server using the aforementioned method. As defined earlier, the probing graph is referred to as $\mathcal{PG}_t$. Given the $\mathcal{PG}_t$, we

utilize the aforementioned features as initial node features $\mathcal{F}_r$. Subsequently, we use GraphSAGE $\mathcal{M}$ to refine these node features through neighbor feature aggregation and use the *lstm* as the aggregation function. The final features $\mathcal{A}_r(u)$ for each node $u$ in this graph are as follows:

$$\mathcal{A}_r(u) = \mathcal{M}_{lstm}(\mathcal{F}_r(u), \mathcal{H}_r(u)) \quad (9)$$

Where $\mathcal{F}_r(u)$ represents the initial features of $u$ and $\mathcal{H}_r(u)$ represents the neighbors' features of node $u$. As the neighboring nodes in this graph provide similar services, we consider that the aggregation operation can enhance the feature representation.

## 3.3 Communication Graph Construction

We gather client IPs, server IPs, and ports from the ISP's environment. Additionally, we also collect the domains of each server from its pointer record (PTR) and the certificate's commonName (CN). These domains have been frequently utilized in server analysis. For each server, we query their certificate from port 443 and *Observed Ports*. We note that despite the certificate being defaultly deployed on port 443, about 18.40% of certificates are deployed on other ports. Therefore, it is necessary to consider the *Observed Ports*. Given an observation time window $t$, we construct a graph $C\mathcal{G}_t$ to describe the communication relationships among servers. Where $\mathcal{L}_i$ represents a node of client IP, server IP, and domain, $h$ denotes the number of nodes, and $\mathcal{R}(\mathcal{L}_i)$ signifies the edges between node $\mathcal{L}_i$ and others. we refer to this graph as the communication graph.

$$C\mathcal{G}_t = < \bigcup_{i=1}^{h} \mathcal{L}_i, \bigcup_{i=1}^{h} \mathcal{R}(\mathcal{L}_i) > \quad (10)$$

We consider that the communication behaviors of VPN servers may be different from that of normal servers. Additionally, previous work [1] has show that topological features are beneficial in graph classification. Therefore we use topological features as the initial node embedding vector in $C\mathcal{G}_t$. The topological features are derived from degree, eigenvector centrality, pagerank, closeness centrality, local clustering coefficient and K-core. These metrics can reveal the intrinsic structural properties and significance of nodes within a network, providing a representation of the server relationships. Considering that these metrics are widely used in graph analysis [27] and the code for constructing these features can be found in public Python libraries [12], we will not introduce their calculation methods. More details can be found in Appendix B.

We use the GraphSAGE $\mathcal{M}$ to capture the features of each node and choose the aggregation function to be *mean*. In this graph, we use $\mathcal{A}_c(v)$ to describe the feature of node $v$. Where $\mathcal{F}_c(v)$ represents the initial features of node $v$ and $\mathcal{H}_c(v)$ represents the neighbors' features of node $v$.

$$\mathcal{A}_c(v) = \mathcal{M}_{mean}(\mathcal{F}_c(v), \mathcal{H}_c(v)) \quad (11)$$

## 3.4 Detection Model

We use the probing graph to represent probing response behaviors and use the communication graph to represent the communication behaviors. For target server $x$, we obtain the features from the probing graph and the communication graph by above methods. Subsequently, we use the formula $\mathcal{Y}(x)$ to enhance their features

and then concatenate all the features together. Where $\mathcal{R}(x)$ represents a linear layer, containing learnable parameters $\mathcal{W}$ and $b$, $\mathcal{R}(x)_*$ denotes a linear layer which contains different $\mathcal{W}$ and $b$. Finally, we use $\mathcal{Z}(x)$ to predict servers.

$$\mathcal{R}(x) = \mathcal{W}x + b \tag{12}$$

$$\mathcal{Y}(x) = \mathcal{R}_1 tanh \mathcal{R}_2(\mathcal{A}_c(x)) \ CONCAT \ \mathcal{R}_3 tanh \mathcal{R}_4(\mathcal{A}_r(x)) \tag{13}$$

$$\mathcal{Z}(x) = Softmax(\mathcal{R}_5 ReLu(\mathcal{R}_6(\mathcal{Y}(x))) \tag{14}$$

## 4 EXPERIMENTS

### 4.1 Dataset

**Offline Dataset** We aim to detect VPN servers in the real world. However, Maghsoudlou et al. [25] disclose that VPN traffic accounts for merely 2.6% in their ISP's environment. There are more than 6.3 million servers in our ISP's environment within one day. Consequently, collecting and labeling VPN servers becomes a significant challenge. Fortunately, previous work [29] has indicated that many VPN vendors deploy their servers in Autonomous Systems (AS) 9009 and 60068. We further obtain these IPs labels from our industry partner **, who claims that their labels are derived from threat intelligence and their detection model. Our partner can identify various server types including VPN servers, Proxy servers, and normal servers. To expedite the data collection process, we regard VPN servers in the aforementioned ASs as seed IPs for dataset collection.

We present a new context-based dataset collection method that assists researchers in collecting VPN servers in the real world. The data collection procedure is as follows: i) Sequence collection: Given a time window $t$, we record the timestamp $\mathcal{T}_i$ when the user $i$ accesses the seed IPs. As mentioned in the background section, clients may access multiple VPN servers within a short time. For user $i$, we gather the server access sequence during the $2\varphi$ seconds $[\mathcal{T}_i - \varphi, \mathcal{T}_i + \varphi]$. ii) Server collection: Since the VPN servers in the same server access sequence may be deployed on different ASs, we can collect many VPN servers from this window. iii) Data Processing: The real-world network environment is highly complex and involves NAT IPs, which may access numerous server IPs within a short time. For instance, we observe that some IPs even access over 1,000 server IPs in just 10 seconds. To eliminate these abnormal sequences, we choose only the server access sequences with a length smaller than $\varrho$.

**Table 2: Offline Dataset Distribution**

| Category | Count | Percent |
|---|---|---|
| VPN | 6840 | Psiphon3 VPN 6.64%, NordVPN 1.09%, Pia VPN 0.92%, Express VPN 0.05%, Surfshark VPN 0.04%, Others 5.20% |
| Proxy | 6295 | 13.54% |
| Normal | 33346 | 71.74% |
| Total | 46481 | 100% |

The distribution of the offline dataset is shown in Table 2. We organize these data into four datasets: VPN and normal servers ($\mathcal{D}_1$), popular VPN and normal servers ($\mathcal{D}_2$), unpopular VPN and normal servers ($\mathcal{D}_3$), and Proxy and normal servers ($\mathcal{D}_4$). The dataset construction method is based on the following considerations: i) There are more than 43 (refer to Appendix, Table 7) different VPN servers

in our dataset. Among them, the servers of the top three VPN vendors (Psiphon3 VPN, Nord VPN, Pia VPN) account for 55.95% of all VPN servers. All of these VPNs adopt private protocols and may pose a challenge to active probing. Particularly, previous work [10] has mentioned that Psiphon3 VPN employ probe-resistant techniques. To this end, we use the $\mathcal{D}_2$ to enable the model to focus on learning features of popular VPNs and to demonstrate the model's identification capability. We use the $\mathcal{D}_3$ to learn features across a multitude of VPNs and to show its generalization capability. Considering the partial functionality similarity between Proxies with VPNs, we also use the $\mathcal{D}_4$ to explore the model's potential detection applicability to Proxies . To our knowledge, there is no publicly available dataset for VPN server identification at the ISP level. Therefore, we make our dataset[2] publicly available to assist the security community in detecting VPN servers.

**Online Dataset** To test the performance of our model in the online environment, we deploy it within the environment of our ISP partner. Different from the offline dataset, we do not specify the seed IPs during the data collection process. We collect 65,636 client access sequences which contain 512,170 server IPs.

### 4.2 Offline Experiments

**Implementation Details** During the data collection stage, we set an observation time window $\varphi$ to 10 seconds, the largest server sequence access length $\varrho$ to 50 and the *Probing Port Combination* similarity $\zeta$ to 0.5 by default. In the subsequent model training stage, the maximum training epoch is set to 150. We choose Adam as the optimizer and set the learning rate to 0.005. Each experiment is conducted 10 times, with the results being averaged for display. To eliminate the impact of data imbalance, we use the downsampling strategy (randomly discarding samples) to ensure consistency between the positive and negative sample data. The models are evaluated by utilizing Accuracy (AC), Precision (PR), Recall (RC), and F1-score (F1). As we have mentioned, servers may not respond with any information after receiving probes. In this situation, we set the corresponding feature value of each model to 0. In our offline dataset, 16.44% of servers do not provide any probing response information.

**Comparison Experiments** Although much work has been done in the field of VPN server detection, the majority of it focuses on traffic information. These methods extract features based on traffic payload, which may be affected by the network environment. In our scenario, we primarily extract features from active probing. Our goal is not to completely replace traffic-based methods, but to demonstrate that not using traffic features can also be feasible. To this end, we only select those methods that are based on active probing for comparison.

The results are shown in Table 3. OOVF [5] focuses on the servers that use the OpenVPN protocol and primarily uses the features related to response time to detect VPN servers. DPP [10] aims to identify servers deploying probe-resistant protocols, principally extracting features from RST/FIN thresholds, response time, and response content. The aforementioned studies have greatly inspired our work. However, their identification approaches primarily focus

---

[2]https://anonymous.4open.science/status/VPNSniffer-1

**Table 3: Offline Experiments**

| Model | $\mathcal{D}_1$: ALL VPN and Normal | | | | $\mathcal{D}_2$: Popular VPN and Normal | | | | $\mathcal{D}_3$: Unpopular VPN and Normal | | | | $\mathcal{D}_4$: Proxy and Normal | | | |
|---|---|---|---|---|---|---|---|---|---|---|---|---|---|---|---|---|
| | AC | PR | RC | F1 | AC | PR | RC | F1 | AC | PR | RC | F1 | AC | PR | RC | F1 |
| OOVF [5] | 0.7793 | 0.8350 | 0.7042 | 0.7641 | 0.8282 | 0.8555 | 0.8018 | 0.8278 | 0.7049 | 0.7856 | 0.5718 | 0.6619 | 0.5369 | 0.6040 | 0.2425 | 0.3460 |
| DPP [10] | 0.6559 | 0.6224 | 0.8018 | 0.7008 | 0.7045 | 0.6517 | 0.8927 | 0.7534 | 0.6284 | 0.6125 | 0.7963 | 0.6924 | 0.7047 | 0.6933 | 0.7903 | 0.7386 |
| ACER [4] | 0.6788 | 0.7042 | 0.6337 | 0.6337 | 0.7052 | 0.6692 | 0.6692 | 0.7530 | 0.7269 | 0.7468 | 0.7074 | 0.7266 | 0.8176 | **0.8118** | 0.8450 | 0.8281 |
| GCN | 0.8353 | 0.8878 | 0.7675 | 0.8233 | 0.8575 | 0.8549 | 0.8612 | 0.8581 | 0.8374 | 0.9119 | 0.7467 | 0.8211 | 0.8071 | 0.7553 | 0.9084 | 0.8248 |
| GAT | 0.8307 | 0.8788 | 0.7671 | 0.8192 | 0.8673 | **0.9616** | 0.7650 | 0.8521 | 0.8358 | 0.8938 | 0.7623 | 0.8229 | 0.8064 | 0.7611 | 0.8932 | 0.8218 |
| GIN | 0.8404 | 0.8337 | 0.8504 | 0.8420 | 0.8451 | 0.7928 | **0.9343** | 0.8577 | 0.8410 | **0.9474** | 0.7222 | 0.8196 | 0.8302 | 0.7735 | 0.9337 | 0.8461 |
| SGC | 0.8333 | **0.9009** | 0.7490 | 0.8180 | 0.8710 | 0.9536 | 0.7798 | 0.8580 | 0.8399 | 0.9156 | 0.7487 | 0.8238 | 0.8180 | 0.7728 | 0.9006 | 0.8318 |
| VPNSniffer | **0.8565** | 0.8382 | **0.8835** | **0.8603** | **0.8932** | 0.8864 | 0.9019 | **0.8941** | **0.8672** | 0.8949 | **0.8323** | **0.8625** | **0.8447** | 0.7915 | **0.9356** | **0.8575** |

**Table 4: Ablation Experiments**

| Method | $\mathcal{D}_1$: ALL VPN and Normal | | | | $\mathcal{D}_2$: Popular VPN and Normal | | | | $\mathcal{D}_3$: Unpopular VPN and Normal | | | |
|---|---|---|---|---|---|---|---|---|---|---|---|---|
| | Accuracy | Precision | Recall | F1 | Accuracy | Precision | Recall | F1 | Accuracy | Precision | Recall | F1 |
| W/G | 0.8399 | 0.8807 | 0.7903 | 0.8330 | 0.8890 | 0.9248 | 0.8508 | 0.8863 | 0.8399 | 0.8803 | 0.7870 | 0.8311 |
| W/PG | 0.5582 | 0.5334 | **0.9288** | 0.6777 | 0.5675 | 0.5411 | 0.8890 | 0.6727 | 0.6279 | 0.5866 | **0.8673** | 0.6999 |
| W/CG | 0.8511 | 0.8492 | **0.8538** | 0.8515 | 0.8529 | 0.9739 | 0.7253 | 0.8314 | 0.8363 | 0.8616 | 0.8616 | 0.8616 |
| W/lstm | 0.8397 | **0.9100** | 0.7539 | 0.8246 | 0.8719 | 0.9774 | 0.7613 | 0.8560 | 0.8235 | 0.8481 | 0.7878 | 0.8169 |
| W/tanh | 0.8531 | 0.8744 | 0.8246 | 0.8488 | 0.8691 | 0.9807 | 0.7530 | 0.8519 | 0.8286 | 0.9209 | 0.7191 | 0.8076 |
| W/MLP | 0.8579 | 0.8685 | 0.8436 | 0.8559 | 0.8626 | 0.9816 | 0.7391 | 0.8433 | 0.8291 | 0.8800 | 0.7623 | 0.8170 |
| Linear | 0.8584 | 0.8636 | 0.8514 | 0.8574 | 0.8641 | 0.8872 | 0.8342 | 0.8599 | 0.8384 | 0.8680 | 0.7984 | 0.8317 |
| CNN | **0.8621** | 0.8973 | 0.8177 | 0.8557 | 0.8774 | 0.9105 | 0.8372 | 0.8723 | 0.8590 | 0.8930 | 0.8158 | 0.8527 |
| LSTM | 0.8246 | 0.8850 | 0.7461 | 0.8096 | 0.8645 | **0.9828** | 0.7419 | 0.8455 | 0.8224 | 0.8903 | 0.7353 | 0.8054 |
| VPNSniffer | 0.8565 | 0.8382 | 0.8835 | **0.8603** | **0.8932** | 0.8864 | **0.9019** | **0.8941** | **0.8672** | **0.8949** | 0.8323 | **0.8625** |

on a limited number of VPNs. This focus may compromise generalization and result in decreased performance on $\mathcal{D}_1$, $\mathcal{D}_2$, and $\mathcal{D}_3$, which include multiple VPNs. Additionally, since OOVF only focuses on the OpenVPN protocol, it has the worst performance on $\mathcal{D}_4$. ACER [4] is designed to detect shadowsocks proxy, so it performs better than OOVF and aforementioned methods in $\mathcal{D}_4$. ACER extracts features from response time, TCP flags, and packet information created during active probing. This method is also adaptable for detecting VPN servers. Additionally, our model can also use GCN [18], GAT [35], GIN [44], SGC [40] to aggregate graph features. We also show their performance in Table 2. The results show that VPNSniffer performs best on all four datasets. Our work proves that it is feasible to use graph-represented behaviors for VPN servers detection.

**Ablation Experiments** As our detection target is VPN servers, we conduct ablation experiments on $\mathcal{D}_1$, $\mathcal{D}_2$ and $\mathcal{D}_3$. The results are displayed in Table 4. To facilitate presentation, we denote the probing graph as $PG$, the communication graph as $CG$, and $W$ signifies 'without'. The results clearly indicate that both $PG$ and $CG$ are beneficial for detection, with $PG$ contributing most significantly. This demonstrates that probing features are more important than topological features. The results of $W/G$ reveal that our model surpasses previous methods by solely utilizing features from response types, response time, response length, termination state, and port distribution. This indicates that our extracted features are more efficient. In the $PG$, we use $lstm$ as the aggregation function in GraphSAGE. The $W/lstm$ means we use the default method $mean$. The better result of $lstm$ might be attributed to the $lstm$ aggregation function's superior information mining capability compared to $mean$. We also conduct ablation experiments about replacing $tanh$ with $relu$ and removing $MLP$ in feature fusion. The results show that all the structures are necessary. Additionally, the results from using a Linear layer, CNN, LSTM as classifiers indicate that our classifier is more suitable.

**Sensitivity Experiments** We perform sensitivity experiments concerning length $\varrho$, time $\varphi$ and similarity $\zeta$ as mentioned in Implementation Details. Figure 6-a illustrates that the shorter the user access sequence length, the higher the F1 value. This may be because shorter sequences contain a higher proportion of VPN servers. In the construction process of the $PG$, we observe that VPN servers might utilize the same port combinations as normal servers, which can lead to misclassifications. When the proportion of VPN servers increases, such misclassifications are reduced. Figure 6-b shows that the model is not sensitive to time $\varphi$. This may be because the data distribution is consistent across different time windows. Figure 6-c indicates that variations in the similarity threshold do not significantly impact the results, possibly because some VPNs have a stable port combination.

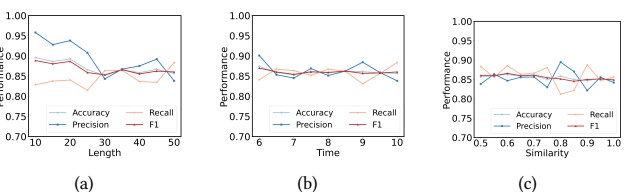

(a)    (b)    (c)

**Figure 6: (a), (b), (c) show the result of Sequence Length, Sequence Time and Similarity, respectively.**

## 4.3 Online Experiments

Our system is designed to assist the security community in detecting VPN servers in the real world. We deploy our system in an ISP's environment and analyze 512,170 servers, identifying 6,143 VPN servers among these. To further evaluate the performance of our model in an industrial environment, we compared it with

**Table 5: Online Experiments**

| Industry Engine | Count | Coverage Ratio |
|---|---|---|
| Ipqualityscore [15] | 4903 | 0.9559 |
| Vpnio [36] | 4405 | 0.8588 |
| IpInfo [14] | 773 | 0.1507 |
| Spur [33] | 3089 | 0.6023 |
| Total | 5129 | 1.0000 |

four industry detection engines. Considering that these detection engines are very expensive, we collect the labels from these detection engines by manually accessing their website. We observe that 83.49% (5,129/6,143) of servers are labeled as VPN by at least one detection engine. Details are shown in Table 5. These results indicate that our system has the potential to enhance the coverage of these advanced detection engines. In real-world detection scenarios, security researchers typically combine multiple methods for detection. VPNSniffer can serve as a supplement to other methods.

Notably, any data we obtain from the detection engine is only used for evaluation purposes, and we guarantee that these data will not be used otherwise. Given our limited dataset, the result may not fully represent the actual capabilities of these detection engines, and our purpose is not to distinguish which is the best.

## 5 DISCUSSION AND LIMITATIONS

**Adaptive attacks** VPNSniffer outperforms other models across various datasets, partially due to the possibility that attackers have altered their tactics in response to previous detection methods. Detection and anti-detection form a continual game of chase. Attackers may refine their strategies to reveal fewer server features after our method becomes public. For instance, they might randomize the RST/FIN threshold, which may require modifying the operating system source code. Attackers can also alter communication behaviors to disrupt the node connection relationships within our graph, but this could compromise user experience. In general, while attackers can employ strategies to bypass our detection, these strategies are likely to incur additional costs.

**Limitation** In this paper, we mainly focus on the features of active probing and topology. We assume that security researchers have the ability to receive probe responses and obtain node connection relationships. However, our method may be limited in some scenarios. If researchers conduct server detection at the gateway, they may not be able to obtain complex node interaction relationships. In this situation, although our method can still be applied, ablation experiments indicate that our model's performance would decline. If researchers conduct detection in situations where internet access is not available, our method is not applicable.

## 6 RELATED WORK

### 6.1 VPN Detection Based on Network Traffic

Numerous work focuses on detecting VPN servers through network traffic. Several studies [3, 6, 26, 39, 41] extract features from various statistical information, such as time intervals, packet lengths, and byte entropy, and then employ machine learning or rule-based methods for detection. For instance, Wu et al. [41] suggest that while servers may randomize traffic to evade detection, researchers

can use byte entropy to identify traffic. Moreover, some research [16, 21, 43, 48, 49] centers on feature extraction from packet length sequences and arrival interval sequences, utilizing deep learning methods for classification. For example, Jiang et al. [16] use sequence information to construct traffic graphs, subsequently employing Graph Neural Networks for identification. Additionally, to avoid manual feature construction, some studies [20, 23] utilize raw packets as features for detection, achieving excellent classification results. Although above work achieve excellent classification results in the training environment, these method may have performance loss due to the changes in environment. For example, Xie et al. [42] demonstrate that some methods might show significant performance degradation in different training networks.

### 6.2 VPN Detection Based on Active Probing

Previous studies have conducted work on active probing for VPN detection, which can be categorized based on the type of protocol. **Traditional Protocols:** Traditional VPN protocols, such as IPsec [13], OpenVPN [5] and PPTP [11] are still widely used[17, 25]. These protocols are open-source and primarily designed to keep communications secure rather than to resist detection. Owing to their open-source and non-resistant nature, researchers can easily construct VPN requests and send probe packets to target servers. By analyzing the response content, they can determine whether the server provides VPN services. Maghsoudlou et al. [25] use this method discover more than 9.8M VPN servers on the Internet. **Probe-Resistant Protocols:** Some studies [10, 45] pointed out that many VPN servers deploy probe-resident protocol, such as Psiphon's obfuscation SSH protocol. These servers reply only to requests that include authentication information. Since it is difficult to acquire authentication information, researchers cannot receive any response after sending probing packets. Frolov et al. [10] observe that probing with popular protocols, such as HTTP, can effectively filter most normal servers out. Additionally, they find that TCP timeout and data thresholds can be used to distinguish VPN servers from others. Xue et al. [45] discover that some commercial OpenVPN server also deploy authentication mechanisms and they utilize the timeout and RST thresholds to detect these OpenVPN servers. There are some studies [4, 8, 37] are not directly aimed at detecting VPN servers but offer methods that could be applied to detect VPN servers. Fifield et al. [8, 37] indicate that HTTP and TLS response may aid in classification. Cheng et al. [4] suggest that researchers could pay attention to metrics such as timeout, TCP flag and response packet length.

## 7 CONCLUSION

In this paper, we introduce a new method for detecting VPN servers using a probing graph and a communication graph. We present some interesting features, such as response types and port distribution, which can help the security community enhance server detection capabilities. Our experimental results demonstrate that using features of active probing and topological relationships is feasible. The results also show that our method outperforms previous methods and can help industrial detection engines enhance coverage ratios.

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

## A  PORT COMBINATION

We show example of some VPNs' Probing Port Combinations in Table 6. Please note that: i) Some VPNs may have multiple port combinations. For instance, Psiphon3 has [443,53,22], [80, 554, 22], [80, 22], and other combinations. One reason for this is that, limited by our observation window (one hour), we do not obtain the complete port combinations, which may be [554,22,53,80,443]. We consider that the larger the observation window is, the more complete port combinations we will obtain. Therefore, researchers can expand their observation window to mitigate this issue. Another reason is that a single VPN may employ different server configuration strategies, resulting in various port combinations. ii) Not all VPNs have stable port combination. Some VPN may use the random port combination.

**Table 6: Example of Probing Port Combinations**

| VPN Name | Probing Port Combination |
|---|---|
| Psiphon3 VPN | {22, 53, 443} |
| Psiphon3 VPN | {22, 443, 554} |
| Psiphon3 VPN | {22, 53, 443, 554} |
| Surfshark VPN | {443, 1443, 4000, 7443, 8443} |
| Pia VPN | {80, 443, 8443} |
| Pia VPN | {80, 443, 8080, 8443, 8888} |
| Hotspot VPN | {80, 443, 563, 636, 993, 995} |
| Hotspot VPN | {80, 443, 563, 636, 993, 995, 6000} |
| Cyber Ghost VPN | {443, 8080, 8081, 8443} |
| Easy VPN | {80, 443, 8080, 8088, 12345} |
| Vpnify VPN | {22} |
| Hide My Ass VPN | {80, 443} |
| Tomato VPN | {22, 80, 443, 8000} |
| Touch VPN | {443, 8082, 8083} |
| Windscribe VPN | {443, 8443} |

**Table 7: VPNs in our dataset**

| VPN Name | VPN Name | VPN Name |
|---|---|---|
| Air VPN | Fastestvpn | Itop Accelerator VPN |
| Browsec VPN | Fastvpnio VPN | Just VPN |
| Cryptostorm VPN | Foxyproxy VPN | Nord VPN |
| Cyber Ghost VPN | Gecko VPN | Opera VPN |
| Daily VPN | Hide My Ass VPN | Pia VPN |
| Deeper Network VPN | Hotspot VPN | Privatevpn |
| Easy VPN | Infvpn | Proton VPN |
| Express VPN | Innovative Connecting VPN | Psiphon3 VPN |
| Witopia VPN | Xvpn | Zenmate VPN |
| Pure VPN | Trust Zone VPN | Thunder VPN |
| Secure Android VPN | Tunnelbear VPN | Tomato VPN |
| Supervpn360 VPN | Turbo VPN | Tor Guard VPN |
| Surfshark VPN | Ultrasurf VPN | Vpn Super Free VPN |
| Vpnify VPN | Vpnunlimited VPN | Windscribe VPN |
| Touch VPN | unknown VPN | |

## B  TOPOLOGICAL FEATURES

Here we describe the calculation methods of topological features. **Degree**: The degree of a node in a graph is the number of edges incident to it. **Eigenvector Centrality**: Eigenvector centrality assigns relative scores to all vertices in the network based on the principle that connections to high-scoring vertices contribute more to the score of the vertex in question than equal connections to low-scoring vertices. **PageRank**: PageRank is an algorithm initially used by Google Search to rank websites in their search engine results. **Closeness Centrality**: Closeness centrality of a node is the reciprocal of the sum of the shortest path distances from the node to all other nodes in the graph. **Local Clustering Coefficient**: The local clustering coefficient of a vertex in a graph quantifies how close its neighbors are to being a complete graph. **K-core**: The k-core of a graph is a maximal subgraph in which every vertex has at least degree $k$. Nodes are peeled off layer by layer, where nodes of degree $< k$ being removed.

