# OpenReview forum: "VPNSniffer: Identifying VPN Servers Through Graph-Represented Behaviors"
_ACM.org/TheWebConf/2024/Conference — TheWebConf24_

### Official Review · Reviewer_aBYp · 2023-11-08

**Novelty:** 4
**Technical Quality:** 4

**Review:**

### Summary
This paper proposes VPNSniffer, a VPN server detection system that utilizes graph-represented probing and communication information to enhance detection performance. VPNSniffer constructs the probing graph and the communication graph, extracts node features from active probing and network traffic, and applies graph neural networks to classify VPN servers. The authors collect and construct new VPN server detection datasets from their ISP partner. In experiments, VPNSniffer can outperform active probing-based methods on offline datasets by a noticeable margin. For online experiments, VPNSniffer can also serve as a complement to existing industry detection engines.
### Pros
1. The idea of constructing and utilizing the probing graph seems interesting and novel.
2. The datasets provided by the authors seem comprehensive and useful.
3. The topic of VPN server detection is significant with a broad range of interests in the web domain.
### Cons
1. The GNN-based VPN server detection methodology is not novel.
2. Missing network traffic-based baselines.
3. Generalization ability claim not justified.
4. No efficiency analysis.
5. Poor clarity. There are many typos, grammatical errors, or inconsistencies throughout the paper.

**Questions:**

1. The authors claim that VPNSniffer has better generalization ability compared to previous methods. But there are no experiments explicitly supporting that claim. For example, The authors may consider an experiment that trains the model on $\mathcal{D}_2$ and evaluates the model on $\mathcal{D}_3$ to verify the claim.
2. There are currently no experimental results of network traffic-based methods. The authors might need to provide some to justify VPNSniffer's superiority to them.
3. VPNSniffer needs to collect information from the ISP network and construct the probing graph and the communication graph, which could cost a considerable amount of computing resources and probing network traffic, especially if the ISP network is large. How is the efficiency of VPNSniffer compared to existing detection methods and industry engines?
4. In online experiments, are all 512,170 servers fed to the industry engines for checking, or only the 6,143 VPNSniffer-identified servers are fed to those engines? It is hard to believe that the VPN servers identified by these industry engines are all within the 6,143 ones identified by VPNSniffer.
5. In the ablation study, there is a variant replacing the classifier with a Linear layer, which is a bit confusing. Isn't VPNSniffer already using a linear classifier, as described in Equation (14)?
6. According to the sensitivity experiments, the performance of VPNSniffer is better when the access sequence length is shorter. Then why do the authors set the length to be 50 for their main experiments, according to line 663?
7. It would be better for the paper to have a dedicated section to formally define the special terms used in this paper, such as "observed port", "probing port", "probing port combination", etc.
8. It would be better for the authors to provide a complete spreadsheet describing all the node features utilized, for both the probing graph and the communication graph.
9. There are many typos, grammatical errors, or inconsistencies throughout the paper. The authors should proofread the paper carefully. Some instances are shown below:
    - Equation (1): why do the authors use summation here? It seems that the summation operation should be changed to the union operation as $s_{(i,*)}^t$ are all sets.
    - Line 316-318: the sentence should be checked again and corrected accordingly. There are some typos and grammatical errors. E.g., "Give" should be "Given", "Poring" should be "Probing".
    - Line 435-436: there should be a "we" before "refer to"
    - Line 555: "show" -> "shown"
    - Inconsistency of notations. The notation of the probing graph is in the calligraphic font ($\mathcal{PG}$) in line 325, but suddenly changed to the normal font ($PG$) in line 737-738. There is a similar issue for the notation of the communication graph.
10. In Section 5, who are the "attackers"? This term first appears in this section and does not make sense.
11. Will the authors release the code implementation of VPNSniffer?

**Ethics Review Description:**

The dataset collected by the authors contains plaintext server IPs, which may be sensitive information with security concerns.

**Ethics Review Flag:**

Yes

**Reviewer Confidence:**

3: The reviewer is confident but not certain that the evaluation is correct

**Scope:**

4: The work is relevant to the Web and to the track, and is of broad interest to the community

---

### Official Review · Reviewer_adPJ · 2023-11-22

**Novelty:** 4
**Technical Quality:** 4

**Review:**

The paper presents a system VPNSniffer which utilizes a graph represented behavior to detect VPN servers in real-world scenarios and presents a comparison outperforming the methods used in 4 offline datasets. Compared to prior works, the authors identify 16.44% more VPN servers through the approach in VPNSniffer which could not be detected through prior approaches used in similar measurements. A large part of the focus of the authors' efforts are in the generalizability and the ability to perform these detections without using the information from the packet payloads. Compared to prior efforts, the VPN datasets in this work present at least 43 VPNs and the deployment at an ISP is a valuable contribution.

This work contributes new observations about stealth ports, presents a new technique with VPNSniffer, and combines features of active probing to node communication relationships. A system like VPNSniffer is intended to be deployed in middleboxes or at ISPs to observe flows and make classifications about the server being contacted.

While the authors mention a small ethics note in their paper it falls short of having a detailed discussion about the challenges and potential mis-uses of the system. The deployment at ISPs to classify VPN gateways/servers with higher accuracy opens up tremendous privacy risks allowing ISPs to learn behaviors about users' VPN usage and could amplify censorship attempts which are rampant on the web today especially by authoritarian governments with a strict control over their ISPs and cellular network services providing Internet access. While no technology is immune to this, the work in this paper is important and worthy of investigation to indicate to the anti-censorship community the ability to overcome detection through this system. As a reviewer, While presenting any mitigation or advice to VPN software developers is out-of-scope of the paper, I'd like the authors to include a more thoughtful ethics section in the paper detailing the potential harms a system such as VPNSniffer could cause which has currently been brushed aside.

The effort in this paper is commendable but some of the formalism presented in the graph construction can be simplified in its writing and can sometimes be hard to follow especially because the reader encounters new terms "Probing port combination", "Observed Port", and additional factors $PL, PC, DT, DP$ which come later. It might be valuable to tabulate the terms used and present the construction as an algorithm keeping it shorter and more focused than being scattered across two pages. This would also leave space for the authors to have a detailed discussion around the attacks against the model, risks of the model, ethical implications and limitations for the work.

**Questions:**

1. Line 105-110 talks about the existing approaches used to determine whether the server is a VPN server based on the response information obtained during the scan. Are these HTTP CONNECT messages or something different? A relatively minor clarification or reference for the statement in Line 110-111 is useful to provide.

2. The VPNSniffer system makes a strong assumption that compared to VPN servers, normal servers are accessed by numerous clients and exhibit a more sophisticated connection relationship (Line 122-124). However with the emergence of public proxies such as the Apple iCloud Private Relay and more architectures which rely on third party proxies, it is hard to argue that the behavior continues to be similar, how does VPNSniffer adapt to these architectural changes in the network affecting the packet flows?

3. What was the anonymization mechanism used to anonymize the IP addresses from the ISP partner? Is it the last octet of an IPv4 address and last X=80 bits of the IPv6 address? Is it hashes of the IP address?

4. Line 350-353 indicates that VPN servers mimic TLS traffic and use port 443 but do not offer TLS services with response behavior being different from normal servers. How do the authors define what normal behavior is? Is this through active probing? Also TLS certificates present on the server might reveal information about the host in the SAN fields of the certificates? How does this work against CDN networks which have millions of requests and possibly to the same set of ports? The approach presented would have high similarity $ζ$ for a large CDN and a VPN service using HTTP CONNECT based mechanism.

5. The classification of DNS response behavior based on connections to port 53 may indicate the existence of an open DNS resolver if $RT_d=1$, but other reasons for errors such as DNS Code 2 (SERVFAIL), or 5 (REFUSED) which return the errors described could mean correctly configured DNS name servers but those which aren't authoritative for the `google.com` query. Did the authors perform reverse PTR queries for the IP addresses to find any associated domains before concluding that the server might be a fake DNS resolver (Line 413). The conclusion seems rather abrupt and could benefit from clarification.

6. Line 600 lists ASNs 9009 (M247) openly advertises VPN capabilities and services from all its global datacenters but 60068 (CDN77) is a CDN network like many others (Akamai, Cloudflare, Fastly -- all of whom have VPN like services). This choice for the seed list potentially might affect the correctness of the classification of the servers.

Minor issues:

1. Line 156, Line 157: should `piracy` instead be `privacy`?
2. Consider Figure 6 to span full width of two columns below Table 4.
3. Line 317, Typo `Poring` --> `Probing`

**Ethics Review Description:**

Authors have an ethics statement but needs more work detailing the immediate harms of such models and tools.

**Reviewer Confidence:**

3: The reviewer is confident but not certain that the evaluation is correct

**Scope:**

3: The work is somewhat relevant to the Web and to the track, and is of narrow interest to a sub-community

---

### Official Review · Reviewer_pCWb · 2023-11-22

**Novelty:** 5
**Technical Quality:** 5

**Review:**

I thank the authors for submitting their work to WWW 24. This paper improves VPN detection rates through constructing communication graphs. The authors primarily leverage two insights: (1) the connection relationship for a normal server is more intricate, and (2) a client might access multiple servers simultaneously. They compare their system with state-of-the-art academic detection systems (in an offline setting) as well as industrial detection engines and demonstrate that their system achieves good performance.

Overall, I really enjoyed reading this paper. It is well-written (modulo various typos), with a solid methodology and systematic evaluation. That being said, below, I list some questions that I have after reading the paper:
* The online evaluation is limited to one dataset (512k server IPs). I wonder if the performance stays stable if the authors sample a subset of IPs from their current dataset. Additionally, I am curious about whether doubling the number of IPs changes the performance (this one might be harder).
* Section 4.3 is currently a bit unsatisfying. I wish the authors could comment on (a) why 16.51% of the servers are not labeled by any industry engine (one way to do this is manually examining a few such servers); (b) how many extra servers are identified by the proposed system; and (c) compared to other industry engines, why IpInfo performs so poorly (e.g., is it because it only labels a specific kind of server?).

Nits:
* Table 1: UCP -> UDP?
* We show their performance in Table 2. It should be Table 3.
* Why 65636? Maybe 65536?

**Post-rebuttal**: I thank the authors for their comments and have no further questions.

**Questions:**

See my reviews above.

**Ethics Review Description:**

Ethical concerns are properly addressed.

**Reviewer Confidence:**

2: The reviewer is willing to defend the evaluation, but it is likely that the reviewer did not understand parts of the paper

**Scope:**

3: The work is somewhat relevant to the Web and to the track, and is of narrow interest to a sub-community

---

### Official Review · Reviewer_K8Lu · 2023-11-22

**Novelty:** 4
**Technical Quality:** 3

**Review:**

This paper proposes a graph-based approach to identify VPN servers. It uses active probe packets for nine application layer protocols (e.g., SSTP, DNS, and HTTP) to elicit responses from the suspicious server machines. From the response packets, the proposed method extracts various features and also construct a graph characterizing communications among different servers. The standard GraphSage technique is used to learn the embeddings of the nodes and thus for VPN server detection. Two types of experiments are performed. For offline experiments, the method is compared against five other existing methods on a labeled dataset contributed by an industry partner. For online experiments, a prototype system is deployed in the partner ISP''s network, showing that the VPN servers detected can be confirmed by existing industry engines.

Strengths:
+ The works considered practical deployment of the proposed system in a real ISP network environment for VPN server detection.
+ The work leveraged various subtle differences in server responses as features to identify VPN servers.

Weaknesses:
- The proposed method is intrusive, as active probe packets are needed. Such packets can also be exploited by a VPN server to hide its existence (e.g., blocking such probing packets).
- The evaluation work is performed on a single dataset contributed by the partnering ISP. It's hard to know whether the techniques can be generalized to other datasets.
- The detection performances do not show consistent improvement over the existing methods.

**Questions:**

* The results presented in Table 3 do not show that VPNSniffer's detection accuracy outperforms the alternative approaches. Particularly, its precision results indicated by precision scores are worse than some other methods. This shows that GraphSage used by VPNSniffer may not be a perfect choice for the detection task.

* Without the ground truth data about the VPN servers detected, it's hard to interpret the detection accuracy results shown in Table 5. Would it be possible that the VPN servers discovered by VPNSniffer could be false alarms?

* The work could be improved by applying the VPNSniffer on other existing VPN datasets. This will help demonstrate the generalization capability of the proposed method.

**Reviewer Confidence:**

2: The reviewer is willing to defend the evaluation, but it is likely that the reviewer did not understand parts of the paper

**Scope:**

4: The work is relevant to the Web and to the track, and is of broad interest to the community

---

### Decision · Program_Chairs · 2024-01-22

**Decision:**

Accept

**Comment:**

The major issues with the paper are related to the data used for testing the solution (lack of generality---a single source (ISP) was used) as well as the intrusiveness of the methods (network probes are needed).
 However, the methodology is sound, the problem interesting, and the approach shows some elements of novelty.
 The interactions between the authors and the reviewers has led to some interesting discussions.

 ---